

# Spatially distributed water-balance and meteorological data from the Wolverton catchment, Sequoia National Park, California

Roger C. Bales[1], Erin M. Stacy[1], Xiande Meng[1], Martha H. Conklin[1], Peter B. Kirchner[2,3], Zeshi Zheng[4]

[1]Sierra Nevada Research Institute, University of California, Merced, CA 95343, USA
[2]Southwest Alaska Network, Inventory & Monitoring, National Park Service, Anchorage, AK 99501 USA
[3]W.A. Franke College of Forestry and Conservation, University of Montana, Missoula, MT 59812 USA
[4]Department of Civil and Environmental Engineering, University of California, Berkeley, CA 94720, USA

*Correspondence to*: Roger Bales (rbales@ucmerced.edu)

**Abstract**

Accurate water-balance measurements in the seasonal, snow-dominated Sierra Nevada are important for forest and downstream water management. However, few sites in the southern Sierra offer detailed records of the spatial and temporal patterns of snowpack and soil-water storage, and the fluxes affecting them, i.e. precipitation as rain and snow, snowmelt, evapotranspiration, and runoff. To explore these stores and fluxes we instrumented the Wolverton basin (2180-2750 m) in

Sequoia National Park with distributed, continuous sensors. This 2006-2016 record of snow depth, soil moisture and soil temperature, and meteorological data quantifies the hydrologic inputs and storage in a mostly undeveloped catchment. Clustered sensors record lateral differences with regards to aspect and canopy cover at approximately 2250 and 2625 m in elevation, where two meteorological stations are installed. Meteorological stations record air temperature, relative humidity, radiation, precipitation, wind speed and direction, and snow depth. Data are available at hourly intervals by water year (1

October - 30 September) in non-proprietary formats from online data repositories (https://doi.org/10.6071/M3S94T).

## 1 Introduction

The western slope of the Sierra Nevada, California, has an elevation-driven climate gradient where wintertime snowpack accumulation provides for a large part of California's annual water needs. Small temperature increases will raise the snowline elevation and alter snowmelt, and subsequent water use by vegetation, streamflow and forest dryness. The



implications of such a shift affect forest management, downstream water management, and ultimately, regional water policy. This research program in the Wolverton basin (1720-3500 m elevation) of Sequoia National Park is part of a larger effort to quantify the two main near-surface stores of water, snowpack and soil-water storage, across the current Sierra Nevada rain-snow transition.

As part of this research, we used distributed snow and soil sensors, as well as remotely sensed data, to characterize snow accumulation and melt, and other components of the distributed water balance. The data set covers late 2006 through the end of water year 2016. This period includes the drought years of 2012-2015, as well as the high snow year in WY 2011.

Kirchner et al. (2014) found that snowmelt at the upper elevations occurs later in the season and at faster rates compared to lower elevations. Historically, the Wolverton basin has been in the seasonal snow zone of the southern Sierra Nevada,

characterized by winter snow accumulation and spring ablation. At the lower elevations, winter snow cover is ephemeral in some years, with 70-95% of the snow-covered season having slow melt rates (Kirchner et al. 2014). At upper elevations, melt occurs during less than 65% of the season. Based on a 2010 set of lidar flights, snow depth peaks at an elevation of 3300 m, which is 200 m below the highest ridge. Between 1850 and 3300 m, snow depth increases at a rate of approximately 15 cm per 100 m; at higher elevations, snow depth decreases rapidly at a rate of 48 cm per 100 m (Kirchner et al. 2014).

Elevation is also the most important factor in under-canopy areas for predicting snow depth variability across the landscape, followed by slope, aspect and canopy penetration fraction (Zheng et al. 2016). The difference between open canopy and under-canopy areas increases through the rain-snow transition up to 25-45 cm at high elevations (Fig. 1).

Meteorological factors, as well as the interference of canopy coverage, impact snow accumulation and melt at these sites. Sub-canopy direct-beam irradiance and a sky-view factor explain the most variation in snow ablation rates, especially at

finer time scales (Musselman et al. 2012). A time-varying canopy parameter in snow modeling reduces errors by seven days in the simulated snow disappearance date, and errors in the timing of soil-water fluxes by 11 days, on average, compared to a bulk parameterization of radiation transfer through the canopy (Kirchner et al. 2014).

Studies of this basin indicate that it is sensitive to a changing climate. The timing and rate of snowmelt indicate this elevation range is sensitive to seasonal meteorology, especially where upper elevations may begin to experience snow melt during

more of the snow-covered season (Kirchner et al. 2014). At both the upper and lower sites, peak soil moisture precedes the average date of snow disappearance, meaning that soil moisture declines even while snowmelt is infiltrating into the soil

system (Harpold et al. 2015). With peak snow depth around 3300 m (Kirchner et al. 2014), such changes to the hydrological

system could have major implications for snow pack water storage and runoff.

## 2 Site Description

The Wolverton basin is a montane, forested catchment northeast of Giant Forest in Sequoia National Park (Fig. 2). In the 5.4

km$^2$ catchment, the dominant aspect is 214° from North, with an average slope of 21.1°, determined by a Lidar flight

(Harpold et al. 2014). Basin coverage is 45-55% forest (Harpold et al. 2014). The forests transition from mixed conifer at the

lower elevations to subalpine forests at the higher elevations. Dominant species are red fir (*Abies manifica*), lodgepole pine

(*Pinus contorta*), western white pine (*Pinus monticola*), incensecedar (*Calocedrus decurrens*), and Jeffrey pine (*Pinus*

*jeffreyi*; Kirchner, 2014). A long, narrow meadow occupies the lower reaches of the western fork, and small meadows are

present in the eastern fork.

Most of the catchment is undeveloped forest with no history of thinning. There is some recreational infrastructure at the base

of the catchment, including trailhead parking lots, a water treatment plant, and recreational buildings. In addition, there are

several recreational hiking trails around the catchment.

During the WY 2007-2016 period, mean annual precipitation was 728 mm at the nearby Lodgepole WRCC weather station.

Mean annual temperature was 6.0°C. The 2008-2010 water years in the Wolverton basin represents dry, normal, and wet

years where peak mean snow depths were 73 - 165% of the 30-year average (Harpold et al 2015). Soil texture in the upper

cm of soil was 6% clay, 17-23% silt, and 71-77% sand (Harpold et al. 2015). Parts of the basin were previously glaciated.

Geology of the basin is granodiorite, granite, and a mix of glacial and alluvial till in the lower reaches (U.S. Geological

Survey map GQ-1636).

## 3 Meteorological data

20

We installed and instrumented two meteorological stations in 2008. The Wolverton meteorological station is at 2180 m, on a

7-m steel triangular-frame tower. The Panther meteorological station, at 2750 m, is a 6.1-m steel pole with cross-arms near

the apex. Sensor instruments and installation heights are listed in Table 1.  Data are recorded at 60-minute intervals, with

control and storage on Campbell Scientific CR1000 datalogger. Program for data acquisition are located on the UC Merced-SNRI digital library with the data.

## 4 Distributed instrument clusters

The distributed sensor nodes were installed in four clusters at the upper and lower elevations in roughly north and south aspects (Fig. 2). The lower-elevation sites, 1 and 2, are south of Wolverton meteorological station and face north and southeast, respectively, at elevations of around 2225 and 2260 m. The upper elevation sites, 3 and 4, are near the Panther meteorological station and face southeast and north, respectively, at elevations of around 2600 and 2640 m. Soil moisture and temperature, snow depth, and meteorological data (air temperature, relative humidity, and solar radiation) are measured at these clusters across the network at hourly intervals. Snow depth is stratified by canopy coverage, with 15 measurements at open canopy location, 6 at the canopy drip edge, and 5 under tree canopies (canopy classification was verified on-site). Distance to snow/soil surface and air temperature are measured with Judd Communications ultrasonic depth sensors, using analog control. Total solar radiation is measured with the LI-COR Environmental LI-200 Pyranometer, like the met stations. Soil volumetric water content (VWC) and soil temperature are measured using Campbell Scientific CS 616 Water Content Reflectometer and 107 temperature probes at depths of 10, 30, and 60 cm below the mineral soil surface. Soil matric potential is measured in one location at each site using Decagon Devices MPS-1 dielectrical water potential sensor at depths of 10, 30, 60, and 90 cm below the mineral soil surface (http://www.decagon.com/products/discontinued-products/mps-1-dielectric-water-potential-sensor/).

Data are recorded at 60-minute intervals, with control and storage on Campbell Scientific CR1000 datalogger, using an AM16/32B multiplexer. Program for data acquisition are located on the UC Merced-SNRI digital library with the data.

## 5 Example data

In water years 2007-2016, snowpack was deeper and more persistent at the higher-elevation sites (2640 m; Fig. 3a). Warmer temperatures at the lower-elevation sites (2245 m; Fig. 3b) result in earlier soil wet-up, higher mid-winter soil moisture storage, and earlier peak soil moisture storage in wet and dry years. Long data gaps at Site 2 in 2013, and at Site 1 in 2016,

are due to battery and level-logger issues. Comparing average (WY 2010), wet (WY 2011), and dry (WY 2012) years, snowpack is persistent at both the upper and lower elevation sites (Fig. 4). However, in WY 2014 and 2015, snow pack receded multiple times during each winter. Soil moisture peaks earlier at the two lower elevation clusters. The Wolverton met station experienced a wider swing in temperatures than the Panther met station. Mean annual temperature was 6.3°C at

the upper met station Panther, and 5.7°C at the lower-elevation Wolverton station, with both warmer maximum and cooler minimum air temperatures at the Wolverton site. Water-level data are available for two locations in the meadow.

## 6 Data availability and structure

Data are available at hourly intervals by water year (WY; October 1 – September 30) from WY 2007 through WY 2016. Data are available through online data repositories. Distributed snow depth, air temperature, and soil moisture and

temperature are available through DOI (10.6071/M3S94T), which links to data storage on a server hosted at University of California, Merced, or directly from our digital library at https://eng.ucmerced.edu/snsjho/files/MHWG/Field/SEKI/Wolverton. These data from the distributed networks have been processed to level 1 (QA/QC) and level 2 (gap-filled, derived) data. Raw data for matric potential, sap flow and solar radiation at each site are also available. Gaps are filled through regression with a nearby sensor node, which is selected based

on the best correlation. Multiple neighboring nodes may be selected if needed, and different neighboring nodes may be used to fill each measurement. Short gaps, or gaps in soil temperature, may be filled through linear interpolation. Process notes (metadata) are available from the data repositories. Data in at each site of clusters (e.g., Site 1 or Site 3) are organized by and named by installation location (P1, P2, etc. with canopy coverage indicated as "DE" for drip edge, "UC" for under canopy, or "O" for open).

## 7 Summary

A ten-year meteorological and hydrologic data record is presented for a catchment in Sequoia National Park, in the southern Sierra Nevada. Distributed snow depth and soil temperature and moisture combined with two meteorological stations and a long-term stream-discharge record provide a means of testing and evaluating hydrologic processes in a productive montane

forest. The Wolverton basin research site serves as a southern comparison point with installations in the Kings River (SSCZO; 2008-ongoing; Bales et al. 2018), Merced River (MRB; 2006-ongoing; Roche, 2017); Stanislaus River (2013-ongoing, Pickard 2015); American River (2014-ongoing; Zhang et al., 2016, 2017a, 2017b); and Feather River (2016-ongoing; Avanzi et al., 2018) basins. Studies compiled to date indicate that hydrological variables, including snow depth, timing and rate of snow melt, and soil moisture, in this part of the Sierra Nevada are susceptible to changing climate patterns like warmer temperature or increased vegetation water demands. We invite others to use this data for their own studies of the basin or as a comparison point to further the discussion.

**Author contributions**

RB, MC and PK designed the sensor networks. PK, ES, XM, and ZZ installed and maintained the sensor networks, and processed the sensor-network data. RB, ES and ZZ prepared the manuscript with contributions from all authors.

**Competing interests**

The authors declare that they have no conflict of interest.

**Special issue statement**

This article is part of the special issue "Hydrometeorological data from mountain and alpine research catchments". It is not associated with a conference.

**Acknowledgements**

We thank Sequoia National Park research and permitting staff, and the staff and research team at the Southern Sierra Critical Zone Observatory. This research was funded in part by NSF EAR-0725097, EAR-1239521, and EAR-1331939 for the Southern Sierra Critical Zone Observatory, and the University of California Merced.



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



Table 1. Meteorological station instrumentation

| Variable | Unit | Wolverton installation height | Panther installation height | Instrument |
|---|---|---|---|---|
| Air temperature | ºC | 6 m | 5.2 m | Campbell Scientific HMP45C |
| Relative humidity | % | 6 m | 5.2 m | Campbell Scientific HMP45C |
| Wind speed, direction | m s$^{-1}$, degrees | 6.8 m | 5.1 m | Campbell Scientific R.M. Young 05103 Wind Speed & Direction Sensor |
| Rain | mm | 5.1 | 4.5 m | Campbell Scientific TE525MM/TE525M Rain Gauge |
| Net radiation | W m$^{-2}$ | 5.7 m | ---- | Kipp & Zonen NR-LITE |
| Solar radiation | W m$^{-2}$ | 4.5 | 5 m | LI-COR LI200 Pyranometer |
| Snow depth, air temperature | cm, ºC | 4.4 | 4.4 m | Judd Sonic Snow Depth Sensor |



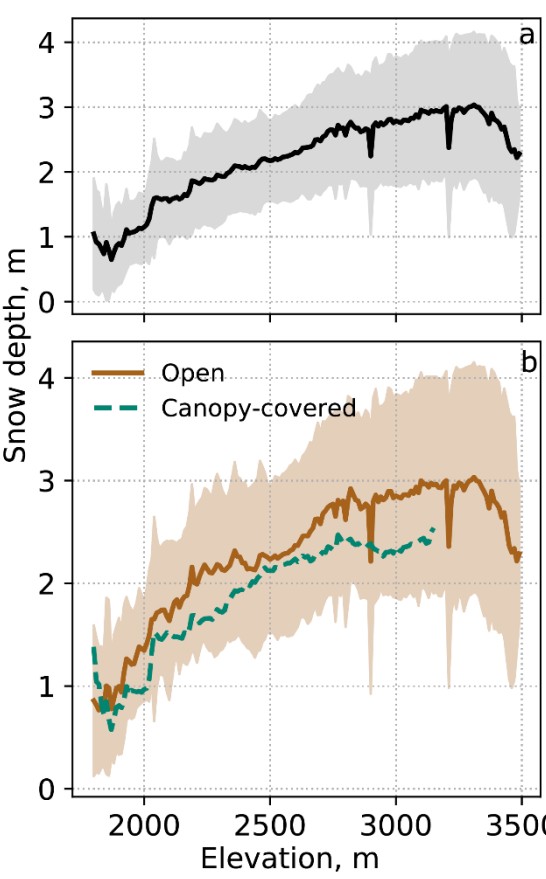

**Figure 1: Snow-depth changes over elevation observed from the lidar data in the Wolverton basin (a) for all snow-depth pixels (b) for snow-depth pixels in the open area and under canopy. On average, snow depth increases with elevation nonlinearly with elevation below 3300 m, decreases when elevation is above 3300 m.**





**Figure 2: Wolverton catchment in the Marble Fork of the Kaweah River, Sequoia and Kings Canyon National Parks. Distributed instruments are clustered around the (a) Wolverton and (b) Panther meteorological stations. (Topographic data: EDNA filled DEM grid, U.S. Geological Survey, 2003. Lidar data: Anderson et al. 2012, snow-off surface raster. Imagery from Digital Globe, (a) 10 October 2015 and (b) 17 September 2016.**



**Figure 3a.** Record of precipitation, soil moisture, snow depth, and air temperature (max and min) at the lower elevation instrument clusters. Near Wolverton meteorological station, Site 1 has a north aspect, and Site 2 has a southeast aspect.





**Figure 3b. Record of precipitation, soil moisture, snow depth, and air temperature (max and min) at the higher elevation instrument clusters. Near Panther meteorological station, Site 3 has a southeast aspect, and Site 4 has a north aspect.**




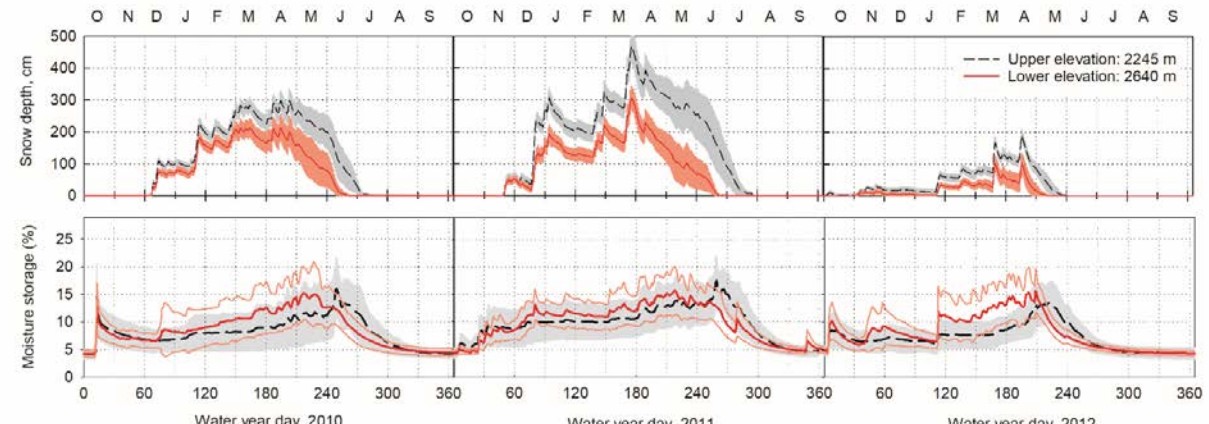

**Figure 4. The 24 sensor nodes highlight the spatial variability of soil moisture and snow depth through Water Years 2010 (average), 2011 (record-setting wet year) and 2012 (extremely dry). Soil moisture can be higher at the lower elevation than the higher elevation, and peak soil moisture may predate the end of snowmelt.**