# Peer review of "Spatially distributed water-balance and meteorological data from the Wolverton catchment, Sequoia National Park, California"

_Earth System Science Data, 2018_

## Referee Comment (RC1) · Anonymous Referee #1 · 7 Aug 2018

This is a very unique set of microclimate data from the Sierra Nevada of California. The data will be of interest to a fairly large group of ecologists, ecosystem scientists and hydrologist working in western mountain catchments. In particular, the soil moisture and temperature data are fascinating and have relevance for understanding the recent large-scale die off of conifers in the Sierra Nevada. I applaud the authors' dedication in installing and operating these stations.

I think the paper and datasets should be accepted after the correction of a few minor technical errors.

Comments on written article: 1. Page 2 line 14. "Rapidly" is vague and unneccesary.

[Figure]

2. Page 3 line 14. Define the acronym WRCC. 3. Page 3 lines 21-24. There are errors somewhere in the elevations of the stations. For example, the text says that the Wolverton met station lies at 2180 m, but on Figure 2, the elevation is given as 2206 m. Similarly, for Panther, the elevation is given as 2750 m in the text, but 2618 m in Figure 2. Please correct. 4. Page 4, first lines. This seems like a sentence fragment. Please improve. 5. Page 4 lines 13-17. Please discuss how the volumetric water content sensors were calibrated. 6. Page 4 lines 18-19. Please state the scan frequency from which the hourly data were computed (5 second, 10 seconds, other?). 7. Pages 4 and 5, Example data section: The section mentions discharge data and level-loggers, but I could find no discharge data in any of the files or in Figure 3. Please add discharge data to the files or else remove discharge measurements from this section. 8. Acknowledgements: The Park name should be Sequoia and Kings Canyon National Parks. 9. Table 1. The instrument column is awkwardly aligned with the other rows. 10. Figure 2. In the uppermost image, there is a blue shaded region that is not defined – is this the region surveyed by LIDAR? 11. Figure 4. In the legend for the figure, the upper elevation, 2245 m, is less than the lower elevation, 2640. This seems like an error.

Level 1 Data Issues 1. In both the Wolverton and Panther met data, there are columns with no data. 2. In the Wolverton data, there is a redundant air temperature column with no data. 3. In the Panther data, what is the difference between average and mean windspeed? The data are identical, so one of the columns should be deleted. 4. At Sites 1 and 2, I noted some negative snow depths. Perhaps an offset could be applied? 5. At Site 3, there seems to be an issue with soil temp_P3_0_10 cm. The values seem too high, relative to the other sensors, for ∼ October 2011 through June 2012. 6. I have attached a Word document with graphs of air temperature and humidity at Wolverton and Panther. Please address the issues raised in the document about site differences.

Please also note the supplement to this comment:

[Figure]

https://www.earth-syst-sci-data-discuss.net/essd-2018-70/essd-2018-70-RC1-supplement.pdf

**Supplement:**

Please comment on the inter-site differences in climate between Wolverton and Panther met stations. In some cases the differences in climate appear to be larger than can be easily explained by differences in elevation and microclimate.

These data indicate that temperatures at Wolverton can be 8 to 12 degrees warmer than at Panther. This difference cannot be explained by the dry adiabatic lapse rate which would predict a max difference of about 5.6 deg C. Could there be any site specific issues with the Wolverton station? Are there structures or parking lots that could cause excess warming?

[Figure]

[Figure]

These data indicate that RH at Wolverton can be substantially higher than at Panther Could there be any site specific issues with the Wolverton station? Are there artificial sources of moisture near the Wolverton station?

Also, note the Panther RH readings from ca. October 2011 and May 2012, where RH>100% which is not possible.

---

## Referee Comment (RC2) · E.H. Bair (Referee) · 22 Sep 2018

General comments:

This is a nice contribution of ten years of hourly model-ready measurements from several high-altitude sites in Seqouia National Park. I appreciate the inclusion of daily and hourly continuous measurements. It's helpful that the authors have provided the Level 0 through Level 2 data to clarify where the data were cleaned and gap filled. I only have a few minor points to suggest, and recommend publication after they are addressed. If the authors have any questions, I encourage them to contact me at nbair@eri.ucsb.edu.

Lidar data are mentioned throughout the text and shown in Figs. 1 & 2, but not provided. I see that the lidar data are provided in Harpold et al. (2014) so I understand that they are accessible via FTP, but the lidar data are not part of this dataset. Fig. 2 is ok because it provides an overview of the site, but I suggest removing Fig. 1.

Other than a few sentences on gap filling, the manuscript does not address many of the potential biases in the measurements. Here are some questions on measurement bias:

1) How was precipitation undercatch from wind accounted for? What type of wind shield was used on the precipitation gauge?

2) According to Campbell Scientific's website, the TE525MM operating range is 0 to 50 deg C, not ideal for measuring snowfall. Thus, I assume a heater was used on this tipping bucket? Heaters cause sublimation when set too high. Likewise, clogging occurs during snowfall when heaters are set too low. Since there were no other tipping buckets in the area, regression cannot be used to fill gaps. Please comment on these potential sources of bias.

3) What are the problems with the net radiation measurements? For example, it's difficult to imagine that the snow surface was perfectly level. A properly leveled radiometer will then be sensing the sun at a different angle than that of the snow surface, leading to erroneous net radiation measurements.

Specifics:

There is a "Net_radiation_ws_correction(W/mˆ2)" column which is not explained anywhere.

The authors should check their headers for misspellings, for example, "Air_teperature..." in "daily_wy2007_2016_wolverton.csv."

The datasets at DASH (DOI: 10.6071/M3S94T) and at the website (https://eng.ucmerced.edu/snsjho/files/MHWG/Field/SEKI/Wolverton) are differ-
ent. For example, the website contains flow data while DASH does not. Programs for the dataloggers are also available on the website, but not at DASH. It seems that the website is more comprehensive. Thus, is it possible to just point the DOI to the website, or mirror the contents of the website at DASH?

L 1, p 5, what are "level-logger issues?" L 2 p 5, "snowpack" and "snow pack" are used. I suggest "snowpack" here and elsewhere.

Table 1 - I suggest adding "m" to the installation height column headers and removing the inconsistently used "m" from each row. "W m^-2" is also truncated under units.

Figure 2 – If the Worldview data were acquired using a NextView license (i.e. free use for federally funded research), then DigitalGlobe has very specific instructions for captions, e.g. "® 2018 DigitalGlobe NextView License."

References:

Harpold, A. A., Guo, Q., Molotch, N., Brooks, P. D., Bales, R., Fernandez-Diaz, J. C., Musselman, K. N., Swetnam, T. L., Kirchner, P., Meadows, M., Flanagan, J., and Lucas, R.: LiDAR-derived snowpack data sets from mixed conifer forests across the Western United States, Water Resour. Res., 50(3), 2749-2755, doi:10.1002/2013WR013935, 2014.

---

## Author Comment (AC1) · 22 Oct 2018

Authors' response to Interactive comments on "Spatially distributed water-balance and meteorological data from the Wolverton catchment, Sequoia National Park, California" by Roger C. Bales et al.

Note: some formatting modifications were made to the referee's comments in the attached file to move each comment to a separate line.

Author response: We appreciate the time and consideration of this review from Anonymous Referee #1. We have responded to individual comments and made changes as

indicated below and in the attached PDF file for this response. With the comments, there is a graph in the attached file that responds to reviewer comments on the data (regarding their supplement).

Anonymous Referee #1 This is a very unique set of microclimate data from the Sierra Nevada of California. The data will be of interest to a fairly large group of ecologists, ecosystem scientists and hydrologist working in western mountain catchments. In particular, the soil moisture and temperature data are fascinating and have relevance for understanding the recent large-scale die off of conifers in the Sierra Nevada. I applaud the authors' dedication in installing and operating these stations. I think the paper and datasets should be accepted after the correction of a few minor technical errors.

Comments on written article: 1. Page 2 line 14. "Rapidly" is vague and unneccesary. Response: Removed the word.

2. Page 3 line 14. Define the acronym WRCC. Response: We removed the acronym and corrected the operator name to the Sequoia and Kings Canyon National Parks. The National Parks are the correct operator, while the Western Regional Climate Center aggregates the data from the full network.

3. Page 3 lines 21-24. There are errors somewhere in the elevations of the stations. For example, the text says that the Wolverton met station lies at 2180 m, but on Figure 2, the elevation is given as 2206 m. Similarly, for Panther, the elevation is given as 2750 m in the text, but 2618 m in Figure 2. Please correct. Response: The elevations included in the text were erroneous; we have corrected these elevations and the text is now consistent with the figure. We also confirmed the elevations for each of the snow depth clusters; these have been corrected to the exact elevations listed in figure 3a and 3b (the elevation of the control box and center of each site) instead of approximations.

4. Page 4, first lines. This seems like a sentence fragment. Please improve. Response: The last two sentences of section 3 (page 4, lines 1-2) were rewritten for clarity.

5. Page 4 lines 13-17. Please discuss how the volumetric water content sensors were calibrated. Response: The Campbell Scientific 616 Water Content Reflectometer sensors for VWC were not specifically calibrated to each soil type at each location, which is one of the flaws with the original experiment design. They were programmed using the standard calibration, which can be further documented using the sensor manual and CR basic program from the sites. Sensor accuracy with the standard calibration is ±2.5%. It is noteworthy that these are all very sandy soils and as such have a pretty narrow range of dielectric constant between zero and saturation for VWC. The standard calibration should be close to accurate and the response pattern is informative. We have added a note to the text on the use of the standard calibration and its accuracy.

6. Page 4 lines 18-19. Please state the scan frequency from which the hourly data were computed (5 second, 10 seconds, other?). Response: The scan frequency is 1 hour. While best practices recommend averaging of multiple scans to compute hourly values, the remote location required less frequent scans for conservation of battery life. We added "scan" after "60-minute" to clarify this in the text.

7. Pages 4 and 5, Example data section: The section mentions discharge data and level-loggers, but I could find no discharge data in any of the files or in Figure 3. Please add discharge data to the files or else remove discharge measurements from this section. Response: Mention of discharge data was removed from section 7, and in section 5, "level-logger" was corrected to "logger".

8. Acknowledgements: The Park name should be Sequoia and Kings Canyon National Parks. Response: Corrected in text.

9. Table 1. The instrument column is awkwardly aligned with the other rows. Response: The vertical justification of that column has been aligned with the other columns.

10. Figure 2. In the uppermost image, there is a blue shaded region that is not defined – is this the region surveyed by LIDAR? Response: Yes, this is the region surveyed

by LIDAR, indicated by the callout line between the LIDAR surface (middle image) and that area on the regional map. That region is now indicated by a black outline with hatching, we added a second callout line to the other end of the region, and updated the legend with the symbology for the lidar extent.

11. Figure 4. In the legend for the figure, the upper elevation, 2245 m, is less than the lower elevation, 2640. This seems like an error.

Response: That error has been corrected in the figure, and the elevations changed to the exact mean elevation of the distributed clusters, rather than an approximation.

Level 1 Data Issues

1. In both the Wolverton and Panther met data, there are columns with no data. Response: The columns that we have in the Wolverton and Panther met data are not empty columns but do have -9999 in Level 1 data record, which indicates that we have the sensors deployed there but the sensors do not function properly. In the README metadata file, we indicate missing readings would be marked by "-9999" in the Level 1 and Level 2 data. We will make an addition to the metadata file that indicates that some sensors are deployed but the data was not processed, as well as the reasons why.

2. In the Wolverton data, there is a redundant air temperature column with no data. Response: This value is from the snow depth sensor and is used to correct the distance calculation for snow depth. There is raw data from this sensor and the corrected data will be added to the Level 1 and Level 2 data files. The variable names files now indicate that the temperature value derives from the snow depth sensor.

3. In the Panther data, what is the difference between average and mean windspeed? The data are identical, so one of the columns should be deleted. Response: The mean windspeed is an intermediate value used to calculate the wind direction. Since it has almost no variation (infrequent differences of less than one-ten-thousandth) from the

measured average windspeed, we will remove this column from Level 1 and Level 2 data.

4. At Sites 1 and 2, I noted some negative snow depths. Perhaps an offset could be applied? Response: Because the baseline is not always stable, some negative values of snow depth persist in Level 1 data after the known offset (from site-visits) is applied. These are addressed in Level 2 data by adjusting the baseline based on the values before the snow season and after snow melt. We will add additional information to the metadata to enumerate the steps taken at each level.

5. At Site 3, there seems to be an issue with soil temp_P3_0_10 cm. The values seem too high, relative to the other sensors, for _ October 2011 through June 2012. Response: Yes, the values for the soil moisture sensors are higher than those at the same depth at P4 and P5 (at the same site), but the moisture wet-up and drying patterns at the sensor in question track well with sensors in the same pit at 30 cm and 60 cm (See Figure 1). There was no obviously erroneous data to remove, and there are pit siting variations that may explain the differences (for instance, P3, or pit at point three, is located closer to boulders and exposed bedrock that may funnel water to the pit, and it may receive more solar energy than P4 as it is further from the tree bole). Instead, the patterns could indicate that sensors at some depths in P4 and P5 may be aging and yield suppressed values, but we have no indication that these are clearly erroneous data. With no obviously erroneous data, we present the data we gathered (with QA/QC for each data processing level).

Figure 1. The soil volumetric water content (VWC) data from points 5, 4, and 3, have been graphed here for depths of 10 cm, 30 cm, and 60 cm.

6. I have attached a Word document with graphs of air temperature and humidity at Wolverton and Panther. Please address the issues raised in the document about site differences.

Response: The evaluation of temperature, instead of showing "temperatures at
Wolverton can be 8 to 12 degrees warmer than at Panther," seems to indicate that Panther Met reads ($\sim$10°C) higher temperatures in the middle of the temperature range than at the same time measurement at Wolverton Met. This variation may be explained considering the placement of each tower. Panther Met is at a sunny, upland area that is generally open, while Wolverton Met is sited at a low spot between a creek and a meadow-like drainage. Cooler temperatures may be expected given the proximity to the water bodies and cold-air drainage effects that Wolverton Met might experience. Regarding the Relative Humidity values, we will correct those erroneous values that were over 100% for Panther Met (when our data manager returns from vacation). The Wolverton Met station is sited between the Wolverton Creek and a small meadow area, which likely increases relative humidity. Cold-air drainage and the temperature-effects of creek proximity could also contribute to the high relative humidity values.

Please also note the supplement to this comment:
https://www.earth-syst-sci-data-discuss.net/essd-2018-70/essd-2018-70-AC1-supplement.pdf
* * *

---

## Author Comment (AC2) · 22 Oct 2018

Authors' response to Interactive comments on "Spatially distributed water-balance and meteorological data from the Wolverton catchment, Sequoia National Park, California" by Roger C. Bales et al.

Author response: We appreciate the time and consideration of this review from Referee #2, Dr. E.H. Bair. We have responded to individual comments and made changes as indicated below, and in the attached file.

E.H. Bair (Referee) nbair@eri.ucsb.edu

[Figure]

General comments: This is a nice contribution of ten years of hourly model-ready measurements from several high-altitude sites in Seqouia National Park. I appreciate the inclusion of daily and hourly continuous measurements. It's helpful that the authors have provided the Level 0 through Level 2 data to clarify where the data were cleaned and gap filled. I only have a few minor points to suggest, and recommend publication after they are addressed. If the authors have any questions, I encourage them to contact me at nbair@eri.ucsb.edu. Lidar data are mentioned throughout the text and shown in Figs. 1 & 2, but not provided. I see that the lidar data are provided in Harpold et al. (2014) so I understand that they are accessible via FTP, but the lidar data are not part of this dataset. Fig. 2 is ok because it provides an overview of the site, but I suggest removing Fig. 1. Response: While the the snow-on and snow-off lidar data available through Open Topography are not currently part of the DOI for the field data, they are relevant to the dataset. Multiple papers using the field measurements for this data have also used the lidar data. The dataset was included in the reference list, and we added citations to the first mention of the dataset in the introduction, as well as to the caption of Figure 1.

Other than a few sentences on gap filling, the manuscript does not address many of the potential biases in the measurements. Here are some questions on measurement bias:

1) How was precipitation undercatch from wind accounted for? What type of wind shield was used on the precipitation gauge? Response: Precipitation records shown in Figures 3a and 3b are from the nearby NOAA cooperative station at Lodgepole, which has a manual, unshielded weighing gauge. We have clarified the source of the data in the figure captions. We have also added text to Section 5 about biases in the precipitation measurements.

2) According to Campbell Scientific's website, the TE525MM operating range is 0 to 50 deg C, not ideal for measuring snowfall. Thus, I assume a heater was used on this tipping bucket? Heaters cause sublimation when set too high. Likewise, clogging

occurs during snowfall when heaters are set too low. Since there were no other tipping buckets in the area, regression cannot be used to fill gaps. Please comment on these potential sources of bias. Response: While a heated precipitation gauge would have been ideal, we were not able to make that installation. As noted above, we have presented precipitation from Lodgepole, which is operated by Sequoia and Kings Canyon National Parks and attended to daily by the rangers. We made changes to captions for Figures 3a and 3b, and to the text in Section 5 about the data.

3) What are the problems with the net radiation measurements? For example, it's difficult to imagine that the snow surface was perfectly level. A properly leveled radiometer will then be sensing the sun at a different angle than that of the snow surface, leading to erroneous net radiation measurements. Response: Net radiation is measured at the met station which is sited on flat ground. We did not measure the evolution of snow surface slope, but have that information for the ground surface and vegetation. We also added the sensors mentioned in Table 1 to the text in Section 3. Specifics:

There is a "Net_radiation_ws_correction(W/mЁĘ2)" column which is not explained anywhere. The authors should check their headers for misspellings, for example, "Air_teperature. . ." in "daily_wy2007_2016_wolverton.csv." Response: Typos in the headers and in the variable names list file have been corrected. The windspeed correction for the net radiation sensor follows guidance from the sensor maker. We have added language to the metadata provided with the data to explain the correction.

The datasets at DASH (DOI: 10.6071/M3S94T) and at the website (https://eng.ucmerced.edu/snsjho/files/MHWG/Field/SEKI/Wolverton) are different. For example, the website contains flow data while DASH does not. Programs for the dataloggers are also available on the website, but not at DASH. It seems that the website is more comprehensive. Thus, is it possible to just point the DOI to the website, or mirror the contents of the website at DASH? Response: There were a few reasons for this decision. The server at DASH has long-term support plans and has a curated set of data with metadata. The website library mentioned here (the Wolverton

folder on the Sierra Nevada San Joaquin Hydrologic Observatory, or SNSJHO) can have unstable links through modification of the file structure; and it has data from multiple graduate students and technicians, including backups of test projects, in some cases with inadequate metadata. Upon review, the flow data have proven questionable and we have decided not to use the data in publications. Thus we did not believe the quality was adequate for inclusion in this published dataset. The well and stream levelogger data could be included as raw data but have not yet had QA/QC conducted for the entirety of the dataset. We will add the datalogger collection programs to the DASH dataset when QA/QC is done.

L 1, p 5, what are "level-logger issues?" L 2 p 5, "snowpack" and "snow pack" are used. I suggest "snowpack" here and elsewhere. Response: "Level-logger" issues should have read simply "logger"; this has been corrected. Snow pack was changed to snowpack here and at the end of the introduction.

Table 1 - I suggest adding "m" to the installation height column headers and removing the inconsistently used "m" from each row. "W mȨ̈E-2" is also truncated under units. Response: The unit "m" was added to the installation height column headers and removed from each row. The table format was adjusted and the "W mȨ̈E-2" unit is now all on one line.

Figure 2 – If the Worldview data were acquired using a NextView license (i.e. free use for federally funded research), then DigitalGlobe has very specific instructions for captions, e.g. "[®] 2018 DigitalGlobe NextView License." Response: The imagery was acquired through Esri's ArcMap, and was cited according to the imagery source at that scale and location. We have updated the citation to full citation recommended by Esri for the entire layer and corrected the "DigitalGlobe" name by removing the space.

References: Harpold, A. A., Guo, Q., Molotch, N., Brooks, P. D., Bales, R., Fernandez-Diaz, J. C., Musselman, K. N., Swetnam, T. L., Kirchner, P., Meadows, M., Flanagan, J., and Lucas, R.: LiDAR-derived snowpack data sets from mixed conifer

forests across the Western United States, Water Resour. Res., 50(3), 2749-2755, doi:10.1002/2013WR013935, 2014.

Please also note the supplement to this comment:
https://www.earth-syst-sci-data-discuss.net/essd-2018-70/essd-2018-70-AC2-supplement.pdf

———————————————